# Fiber Characteristics and Mechanical Properties of *Oxytenanthera abyssinica*

**DOI:** 10.3390/plants12162987

**Published:** 2023-08-18

**Authors:** Linpeng Yu, Fukuan Dai, Kangjian Zhang, Zehui Jiang, Mingsong Xia, Youhong Wang, Genlin Tian

**Affiliations:** 1Institute of New Bamboo and Rattan Based Biomaterials, International Center for Bamboo and Rattan, Beijing 100102, China; yulinpeng2023@163.com (L.Y.); daifuk@163.com (F.D.); jiangzehui@icbr.ac.cn (Z.J.);; 2Key Laboratory of National Forestry and Grassland Administration/Beijing for Bamboo & Rattan Science and Technology, Beijing 100102, China; 3School of Forestry and Landscape Architecture, Anhui Agricultural University, Hefei 230036, China; fliermouse2@126.com (K.Z.); wangyh@ahau.edu.cn (Y.W.)

**Keywords:** *Oxytenanthera abyssinica*, fiber, nanoindentation, cellulose crystallinity, microfibril angle

## Abstract

Unlike the culm hollow structure of most bamboo species, *Oxytenanthera abyssinica* has a unique solid or semi-solid culm, which may endow it with superior mechanical performance. In this study, the variation in fiber morphology and micro-mechanical properties across the radial regions of bamboo culm was examined by optical microscopy, scanning electron microscopy, X-ray diffraction, and nanoindentation. Results showed that the mean values of vascular bundle frequency and fiber tissue proportion were 1.76 pcs/mm${}^{2}$ and 21.04%, respectively, both of which increased gradually from inner to outer. The mean length, diameter, and length-diameter ratio of the fiber were 2.10 mm, 21.54 μm, and 101.41 respectively. The mean indentation modulus of elasticity (IMOE) and hardness were 21.34 GPa and 545.88 MPa. The IMOE exhibited a significant increase from the inner to the middle region, and little change was observed from the middle to the outer region. There were slight fluctuations in hardness along the radial direction. The mean crystallinity and microfibril angle(MFA) of the fibers was 68.12% and 11.26 degrees, respectively. There is a positive correlation between cellulose crystallinity and the IMOE and hardness, while there is a negative correlation between the MFA and the IMOE and the hardness.

## 1. Introduction

Within the plant kingdom, bamboo exhibits a distinctive architecture that bears resemblance to a unidirectional, fiber-reinforced composite [1]. Bamboo also represents a renewable forest resource with a long history of cultivation and utilization around the world, and its efficient utilization and management will benefit the developments in economic, ecological, and social domains [2]. Moreover, bamboo’s growth rate is remarkably fast, and even its maturity can be achieved within a single year [3]. It also has a three-fold greater carbon absorption capability than other plants, absorbing up to 3.7 m${}^{3}$ CO${}_{2}$ per hectare and day [4]. These unique properties of bamboo make it a promising material for a wide variety of applications, such as construction, furniture, papermaking, and so on [5]. *Oxytenanthera abyssinica*, an indigenous bamboo species in Ethiopia and having an exclusive characteristic of tropical Africa, belongs to the subfamily *Bambusoideae* and the family *Poaceae*. It accounts for 85% of the approximately one million hectares area of pure natural bamboo forest in Ethiopia [6,7]. *O. abyssinica* can grow to a height of 13 m and a diameter of 10 cm, which is to be largely used for construction, fencing, furniture, and fuel [8]. Unlike the hollow structure of bamboo culm among most of the species, *O. abyssinica* has a solid or semi-solid culm, which is regarded as a highly distinctive trait and might possess superior mechanical performance. The distinctive macroscopic properties of bamboo are largely linked to the graded distribution of fibers in its ground tissues, where fibers act as the reinforcing phase and play a crucial role in the load-bearing capacity of a material [9].

The use of natural plant fibers has undeniably played a significant role in driving economic prosperity and sustainability in modern society. Among them, bamboo fibers hold a prominent position and have been used in a wide range of industrial applications such as textiles, paper, and construction, etc. [10,11]. Bamboo fibers are an outstanding raw material to produce paper [12]. Bamboo paper has a smooth surface as well as stable mechanical strength, in addition to high color saturation and glossiness [13]. The optimal mixing ratio (50:50 *w*/*w*%) of bamboo- and kenaf fibers has been shown to enhance both the dimensional stability and dynamic mechanical properties of hybrid composites, making them suitable for high-demand applications in the building materials industry [14]; Moreover, the growing interest in employing bamboo fibers as reinforcement agents in polymer composites is primarily attributed to their unique combination of low density, cost-effectiveness, biodegradability, and excellent mechanical properties, which make them an attractive alternative to other reinforcing materials [15,16]. The mechanical properties of bamboo fibers, specifically their structural strength and stiffness are generally determined by a combination of contributing factors, including fiber morphology, cellulose crystallinity, and microfibril angle (MFA). The geometric shape of fibers, including diameter and length-diameter ratio indices, is one of the important factors affecting their mechanical properties [17]. Crystallinity analysis is crucial for practical purposes as it helps explain many physical properties of fibers [18]. As emphasized by other researchers, the degree of MFA exerts a critical impact on the mechanical properties of fibers, thus representing a key structural parameter as well [19,20]. By conducting tensile tests of individual fibers, it was observed that fibers possessing smaller MFA exhibit superior strength and stiffness in contrast to fibers with larger MFA [21]. The application of nano-indentation technology in the micromechanics of wood has facilitated the direct characterization of the elastic modulus, hardness, and viscoelastic properties of wood cell walls at sub-micrometer levels. This technique has also been employed for the characterization of bamboo fibers [22,23]. The solid bamboo species *O. abyssinica*, which possesses exceptional characteristics, has received much less attention to the morphology and micromechanics of its fibers. In this study, various methods were used to examine the microstructure of bamboo culms, including the tissue proportion, dimensions, crystallinity, and mean microfibril angle of fibers, and the frequency of vascular bundles, which are the main structural units of bamboo. The micromechanical properties of fibers, which are the main load-bearing components of vascular bundles, were also measured by using a Triboidenter device. By delving deeper into fiber morphology and composition, a deeper understanding of the underlying mechanisms behind the exceptional mechanical properties of bamboo can be achieved. This knowledge laid the foundation for the development of advanced biomimetic materials with extraordinary mechanical performance.

## 2. Materials and Methods

### 2.1. Materials

Three healthy four-year-old *Oxytenanthera abyssinica* were collected in Ethiopia with 11.5% water content, which was randomly selected. The internode (length 32.86 ± 2.84 cm, diameter 40.11 ± 1.80 mm) at 1-m was then chosen as the sample. The radial sections were divided into three parts for further analysis, namely the inner, middle, and outer regions (Figure 1).

### 2.2. SEM

Small pieces of samples (5 mm × 5 mm × 5 mm) were cut from three different regions, respectively, and then immersed in water and microwaved for 15 min (G80W23CSP-Z, Galanz, Foshan, China). The cross-section of samples was polished by a sliding microtome (Leica SM2010R, Leica, Germany). Prior to imaging, the samples were coated with gold using a sputter coater (E-1010, Quorum Technologies, Emsworth, UK) for 90 s. The polished samples were then imaged by SEM (GeminiSEM 360, Carl Zeiss, Germany) at an acceleration voltage of 3 kV.

### 2.3. Obtaining Frequency and Tissue Proportion

Clear visible cross-sections of vascular bundles and parenchymal cells were obtained using a Buehler IsoMet4000 precision cutting saw (Buehler, Lake Bluff, IL, USA). The cross-sectional areas of the bamboo were scanned using an Epson perfection V850 pro-high-resolution scanner (Seiko Epson Corporation, Shenzhen, China) at a resolution of 9600 pixels per inch (PPI) in 16-bit grayscale mode. The frequency of vascular bundles was obtained by analyzing the cross-sectional images with ImageJ software (National Center for Biotechnology Information, version 1.53 e, Bethesda, MD, USA) [24]. The vascular bundles within the delimited area were counted and their density per square millimeter was calculated. In addition, the area occupied by the fiber within the delimited area was measured and its ratio to the delimited area was used as the tissue proportion of the fiber, more than 30 measurements per data set.

### 2.4. Maceration

Matchstick-sized samples were immersed in the mixture consisting of 30% hydrogen peroxide and glacial acetic acid at a 50:50 weight ratio. Analytical grade glacial acetic acid and 30% H${}_{2}$O${}_{2}$ were purchased from Nanjing Chemical Reagent Co., Ltd. The reaction took place at 60 °C for approximately 8 h using an oven (DHG-9240A, Shanghai Yiheng Technology Co., Ltd., Shanghai, China) to soften the material. Subsequently, fibers were washed with deionized water and then separated. A droplet of the fiber suspension was placed on a microscope slide for observation using a microscope (Leica Microsystem, Biberach, Germany). The length and diameter of thirty randomly selected intact fibers from each region were measured based on fiber micrographs.

### 2.5. X-ray Diffraction (XRD)

The bamboo chips were pulverized to pass through a 60-mesh sieve. The bamboo powders were suspended in distilled water and vigorously stirred. After standing for 5 min, the difference in density caused the fibers to precipitate while parenchyma cells (PCs) floated, so fibers and PCs were separated from each other. The fibers were collected from the bottom of the beaker and were dried for XRD measurement (XD6, Persee, Beijing, China). X-ray diffraction patterns of the fiber samples were measured with continuous scanning over a 2θ angular range of 5°–45°, a step size of 0.02°, and a count time of 50 s per step. The crystallinity index (CrI) was calculated using the formula [25]: (1)$$CrI=\left(I_{200}-I_{am}\right)/I_{200}*100\%,$$
where $I_{200}$ represents the maximum intensity of the peak at approximately 2θ = 22.5°, and $I_{am}$ represents the minimum intensity between the (200) and (100) peaks, evaluated as the peak of the amorphous portion at around 2θ = 18°. More than 30 measurements per data set. The Origin 2018 software was used to process the obtained raw data using the aforementioned method.

Fiber bundles were stripped from the bamboo strips by hand and then were arranged closely together to form opaque sheets with a dimension of 1 mm in thickness, 10 mm in width, and 15 mm in length. The tangential section of the fibers was scanned by a point-focused X-ray (XD-2/XD-3, Persee, Beijing, China) beam with Cu Kα radiation (λ = 1.54060 Å), generated with a voltage of 40 kV and a current of 30 mA. The diffraction pattern analyzed the cellulose crystalline structure, with measurements taken at a position of 2θ = 22.4°, rotating the sample around its normal axis at a rate of approximately 8° per minute [26]. The mean MFA is calculated based on a method developed by Cave [27], which involves drawing a tangent at the point of inflection and using the T parameter as an angle indicator. The MFA is then obtained using the Cave equation, for which a nonlinear least-squares routine is utilized to fit angle-intensity data to a Gaussian curve, effectively removing diffraction peak noise. MFA is determined from the intensity peak with the strongest Gaussian fit, which in this study corresponds to peaks observed at 2θ≈ 88° and 268°, and used to calculate the T parameter. More than 30 measurements per data set. The Origin 2018 software was used to process the obtained raw data using the aforementioned method.

### 2.6. Nanoindentation Test

The samples were cut into a conical shape with the fiber of interest located at the tip of the cone. The tip of the cone was polished using a diamond blade. A Triboindenter (Hysitron, Minneapolis, MN, USA) was used as the test apparatus. A Triboindenter as the test apparatus has more cost-efficiency and less time-consuming data analysis compared to alternative single molecule techniques [28]. The fiber of interest was subjected to a three-stage load in force control mode. The measurement began with a loading rate of 50 μN·$s^{-1}$ for 5 s until the maximum load of 250 μN was achieved, and then kept the maximum load for 6 s to minimize experimental errors caused by viscoelastic materials. After completing a 3 s unloading segment, data from the unloading segment was assessed for computational analysis (Figure 2). The indentation modulus of elasticity and hardness of the samples were then calculated using the Oliver and Pharr equations [29], incorporating specific data and procedures as described by Eder et al. [30] and Dai et al. [31,32]:(2)$$1/E_{r}=\left(1-v^{2}\right)/E-\left(1-v_{i}^{2}\right)/E_{i}$$
(3)$$H=P/A_{c}$$

The indentation hardness (*H*) is calculated using the peak load (*P*) and the projected area at peak load ($A_{c}$). The composite response modulus ($E_{r}$) can be determined by analyzing the load-displacement curve and utilizing elastic contact theory. The elastic modulus ($E_{i}$) and Poisson ratio ($v_{i}$) of the tip are known value for diamond tips, with ($E_{i}$) equal to 1141 GPa and vi equal to 0.07. The indentation modulus of elasticity (*E*) and Poisson ratio (*v*) of the sample is also required for calculations, with the values previously determined to be 0.22 and reported by Yu et al. [33] and Xing et al. [34]. The sample surfaces were scanned before and after testing to select valid nanoindentation test data for analysis of the results, where three samples had more than 30 valid data.

### 2.7. Data Analysis

Data was measured from the obtained images using ImageJ. Statistical analysis of the data was performed using Excel 2021, ANOVA analysis was conducted using IBM SPSS Statistics 19 software, and charts and diagrams were created using Origin 2018 software and Powerpoint software.

## 3. Results and Discussion

### 3.1. Structure of Cross Section

The cross-section of *O. abyssinica* in Figure 3 reveals that the bamboo column is predominantly composed of PCs that were embedded with vascular bundles. The middle vascular bundle belongs to the broken waist type (Figure 3c), while the inner and outer vascular bundles belong to the semi-open type (Figure 3a,e). The fact that the vascular bundles are embedded in the PCs and unevenly distributed along the radial direction highlights the gradient variation nature of bamboo. The primary structural components of vascular bundles are hollow metaxylem vessels (MV) and fiber caps/sheaths made of sclerenchyma tissue. The fiber bundles that form a protective layer around the vascular bundles occurred more frequently in the outer region than in the inner region. Although vascular bundles occurring in the middle region exhibit a lower frequency than in the outer region, these middle fiber bundles exhibit a larger size. The fiber caps/sheaths consist of many longitudinal single bamboo fibers. The chemical composition of bamboo fiber includes cellulose (26.0–43.0%), hemicellulose (20.5%), lignin (21.0–31.0%), pectin (negligible content), wax (negligible content), ash and moisture (negligible content) [35]. Figure 3b,d,f illustrates the microstructure of an individual bamboo fiber, which is characterized by a concentrically layered wall structure, the layers consist of a thick cell wall surrounding a small lumen. Among the three regions, the outer fibers have thick cell walls and small lumens, which can be viewed as a structural adaptation that facilitates the attainment of high bending stiffness and strength [36].

### 3.2. The Frequency of Vascular Bundle and the Tissue Proportion of Fiber

The vascular bundle frequency of inner, middle, and outer were 0.59±0.13 pcs/mm${}^{2}$, 1.19±0.17 pcs/mm${}^{2}$, and 3.50±0.19 pcs/mm${}^{2}$, respectively. The fiber tissue proportion of inner, middle, and outer were 3.78±0.98%, 26.01±1.41%, and 33.33±4.94%, respectively. This indicated that both the vascular bundle frequency and the fiber tissue proportion increased radially from inner to outer (Figure 4). The vascular bundle frequency and the fiber tissue proportion of the inner was much lower than those of the middle and outer. Vascular bundles were the main structures responsible for the longitudinal transport of water and nutrients in bamboo, as well as the source of high mechanical strength, toughness, and bending resistance of bamboo material [37,38]. This implied that in *O. abyssinica*, the solid part had poor transport and load-bearing functions. Although the frequency of vascular bundles in the outer region was 2.94 times higher than that in the middle region, their fiber tissue proportion was similar. This was because the vascular bundles in the middle region had larger dimensions and more fiber content, which agreed with the characteristics of vascular bundles observed under electron microscopy.

### 3.3. Fiber Dimensions

In bamboo, the dimension of fibers influenced Young’s modulus, tensile strength, flexural properties, and crush resistance of bamboo. These properties were important for evaluating the performance of bamboo as a structural material [39]. Moreover, fiber dimension also affected the surface roughness and wettability of bamboo fibers, which could influence their interfacial adhesion and bonding with adhesives [40]. Therefore, fiber dimension was a key parameter that needs to be considered when assessing the quality and potential of bamboo fibers for different applications. The length of fibers from inner to outer was 1.71±0.55 mm, 2.30±0.51 mm, and 2.28±0.62 mm, respectively (Figure 5), where the inner one was the shortest. After one-way ANOVA statistical analysis, significant differences in fiber length were found between the inner and the middle, as well as between the inner and the outer. However, there was no significant difference between the middle and the outer. The mean fiber length of *O. abyssinica* (2.10 mm) is close to that of *Phyllostachys edulis* (2.00 mm) and *Bambusa emeiensis* (2.20 mm) [41], but shorter than that of *Dendrocalamus giganteus Munro* (2.72 mm) [42]. Based on the frequency distribution of fiber length, it can be inferred that the fibers from *O. abyssinica* belong to the category of long-fiber raw materials, which can be considered a high-quality pulp material for producing superior paper products. Similarly, composite materials made of long bamboo fibers have high tensile and bending strength, suitable for construction, transportation, and other fields [43].

The diameters of the fibers were 20.09±4.95 μm, 24.03±5.69 μm, and 20.51±6.00 μm from inner to outer, fibers in the inner region have the smallest diameter and fibers in the middle are the largest (Figure 5). After one-way ANOVA analysis, there was no significant difference in the diameter of the fibers along the radial direction. *O. abyssinica* fibers have a relatively uniform diameter along the radial direction, and considering the difference in length, the fibers in the inner region appear to be shorter and thicker than those in other regions. The fiber diameter in this study was close to that of *Gigantochloa levis*, or *Gigantochloa scortechinii* [44].

The length-diameter ratios of the fibers from the inner to outer regions were 87.15±26.58, 98.64±22.67, and 118.45±41.07, respectively. The observed increasing trend in the length-diameter ratio from the inner to outer regions suggests that the fibers in the outer regions are more elongated and slenderer than those in the inner regions (Figure 5). After one-way ANOVA analysis, there was no significant difference in the length-diameter ratio of the fibers along the radial direction. The size of the length-diameter ratio of bamboo fibers is an important factor that affects the quality of paper [45]. The greater the length-diameter ratio of the fiber, the higher the tear strength of the paper, whereas fibers with a low length-to-diameter ratio below 45 exhibits less suitability as raw materials in the papermaking process [46]. The length of the most widely distributed bamboo fiber of *Phyllostachys edulis* ranges between 1.23 and 2.71 mm, which satisfies the demands of most paper types [47]. In contrast, fibers from *O. abyssinica* are between 1.71 and 2.30 mm in length and have a smaller size distribution range, which makes them more suitable as a raw material for paper production [48].

### 3.4. Cellulose Crystallinity and Microfiber Angle

The mechanical properties of fibers are influenced by factors such as their crystallinity and MFA [49]. The CrI was derived from the data obtained by calculation(Figure 6). The cellulose crystallinity of the fibers in the inner, middle, and outer regions was 67.81±1.94%, 68.65±1.32%, and 67.91±0.72%, respectively (Figure 7). While the cellulose crystallinity of the middle fibers exhibited the highest value, the difference between inner and outer fibers was found to be negligible. One-way ANOVA analysis showed that there was no significant difference in the cellulose crystallinity of the fibers along the radial direction. The fibers from *O. abyssinica* demonstrated a higher degree of crystallinity compared to *Phyllostachys edulis* and *Dendrocalamus farinosus*. As cellulose crystallinity is known to reflect the physical properties of bamboo fibers to a certain extent [50], suggesting that *O. abyssinica* may possess superior mechanical properties.

The inner, middle, and outer regions in *O. abyssinica* demonstrate mean MFA of 11.72±0.88, 10.52±0.75, and 11.54±0.77 degrees, respectively, in the radial sections (Figure 7). The one-way ANOVA analysis indicates that there is no significant difference in the MFA of the fibers along the radial direction. In general, there is a negative correlation between MFA and mechanical strength, which is due to the fact that the smaller the MFA, the more parallel the cellulose molecules are to the cell axis, which can increase the stiffness and tensile resistance of the cell wall, and thus improve the overall elastic modulus and tensile strength [51,52].

### 3.5. The Mechanical Properties of Fibers

Nanoindentation is currently one of the most-used methods to characterize the mechanical properties of various materials at submicro- and nanoscale levels. The trend of mechanical properties of the fiber cell wall from the inner to the outer is clearly illustrated in Figure 8. The indentation modulus of elasticity (IMOE) from inner to outer was 19.08±1.20 GPa, 22.79±1.11 GPa, and 22.14±1.35 GPa, respectively. The hardness from the inner to the outer was 514.79±62.85 MPa, 558.21±33.05 MPa, and 564.65±47.46 MPa, respectively. The inner region exhibited the smallest values for both the IMOE and the hardness. In comparison, there was a significant increase in the indentation modulus of elasticity in the middle region as compared to the inner region, but the outer region showed little change in the indentation modulus of elasticity compared to the middle region. One-way ANOVA analysis showed the difference in IMOE between the inner and middle regions was found to be extremely significant, while the IMOE difference between the middle and outer regions was not significant. The hardness of the inner region was 3.71 GPa lower than that of the middle region, while the hardness of the outer region was 0.65 lower than that of the middle region. The middle region exhibited the greatest fiber hardness, but there were only minor fluctuations in hardness along the radial direction, which did not result in significant differences. Overall, the strength of cell walls remained stable from the inner to the outer regions.

The mean IMOE and hardness of fibers in *O. abyssinica* was 21.34 GPa and 545.88 MPa, respectively. The cell walls of *O. abyssinica* exhibit superior mechanical properties that are comparable to the widely used fibers of *Phyllostachys edulis* (21.7 GPa and 610.00 MPa) but demonstrate superior strength when compared to *Dendrocalamus farinosus* (18.56 GPa and 410.72 MPa) [53,54]. This indicates the fibers of *O. abyssinica* may have a higher resistance to deformation and fracture than *Dendrocalamus farinosus*. However, the distinctive non-hollow structure of *O. abyssinica* implies that its fibers may be a more robust and reliable choice for use in various applications in which high tensile strength and structural integrity are demanded. The cellulose crystallinity exhibits a positive correlation with the indentation modulus and hardness, indicating that highly crystalline fiber has higher mechanical strength and better resistance to deformation. In contrast, a negative correlation exists between the MFA of fiber and the indentation modulus and hardness, suggesting that a lower MFA led to higher mechanical strength and better resistance to deformation. Therefore, these mechanical properties of fibers are closely associated with fiber structural characteristics, which can potentially guide the optimization of the processing and synthesis of fiber materials with desired mechanical properties.

The micromechanical properties obtained by nanoindentation are usually much higher than the macroscopic mechanical properties, because the macroscopic mechanical properties are the result of the coupling of vascular bundles with higher mechanical properties and parenchyma tissues with lower mechanical properties, while the target of nanoindentation is the fiber cell wall with better mechanical properties [55]. This indicates that the macroscopic mechanical properties are mainly influenced by the micromechanical properties and the tissue proportion of fibers. The fibers of *O. abyssinica* not only have a high tissue proportion but also have quite superior micromechanical properties, which make the macroscopic mechanical properties of *O. abyssinica* possibly better than some common bamboo.

## 4. Conclusions

In *O. abyssinica*, the vascular bundle frequency and the fiber tissue proportion increased radially from inner to outer. The vascular bundle frequency and the fiber tissue proportion of the inner were much lower than those of the middle and outer. The inner fibers were found to be the shortest, while there was no significant difference in length between the middle and outer fibers. The diameter of fibers did not vary significantly across radial regions, but the length-diameter ratio increased from the inner to outer sections.

There is no significant difference in cellulose crystallinity and the MFA among the inner, middle, and outer fibers from *O. abyssinica*. The indentation modulus of elasticity exhibited a significant increase from the inner to middle region, However, little change was observed from the middle to outer region. Although there were minor fluctuations in hardness along the radial direction, they did not result in any significant differences. The cell wall strength of the bamboo fiber remained stable from the inner to outer regions. The results indicate that there is a positive correlation between the cellulose crystallinity and the indentation modulus of elasticity and hardness, while there is a negative correlation between the MFA and the IMOE and hardness.

The microstructure of *O. abyssinica* reveals some fascinating features, but there are still many unanswered questions about this material. One of the key areas that require further investigation is the mechanical properties of *O. abyssinica*. How do Young’s modulus, tensile strength, flexural properties, and crush resistance of *O. abyssinica* fibers compared with other bamboo or plant fibers? How do these properties affect the potential applications of *O. abyssinica* as a biomaterial? Another important aspect that needs to be studied is the wettability properties and compatibility of *O. abyssinica* surfaces with different adhesives. Interfacial adhesion mechanisms are among the most influential yet seldom discussed factors that affect the physical, mechanical, and thermal properties of plant-fiber-reinforced polymer composites. In particular, the chemical nature of the fibers and the matrix can greatly affect the interfacial adhesion between them. Bamboo fibers are highly lignified, similar to kenaf and jute fibers. Lignin is a hydrophobic polymer, which means that it does not interact well with polar matrices such as polybutylene succinate (PBS). On the other hand, lignified plant fibers can display strong interfacial forces with non-polar matrices such as polypropylene (PP) [56]. These factors could have a significant impact on the performance and durability of biocomposites made from this material.

## Figures and Tables

**Figure 1 plants-12-02987-f001:**
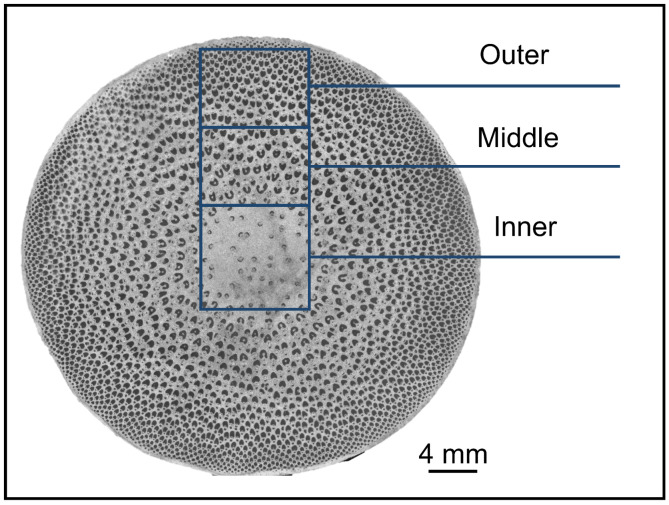
Representative schamatic representation of *O. abyssinica*.

**Figure 2 plants-12-02987-f002:**
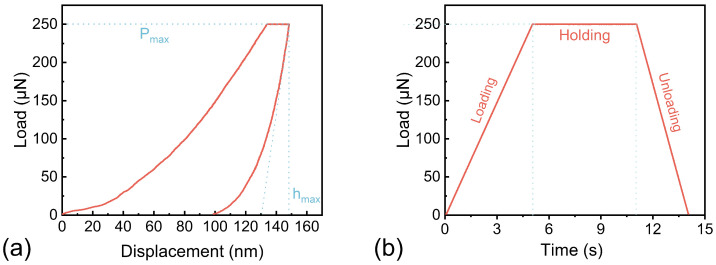
(**a**) The control schematic for Load-Time comprises of loading section, loading holding section, and unloading section, presented in sequence. (**b**) The standard force-displacement curves correspond to the three loading processes visualized in (**a**).

**Figure 3 plants-12-02987-f003:**
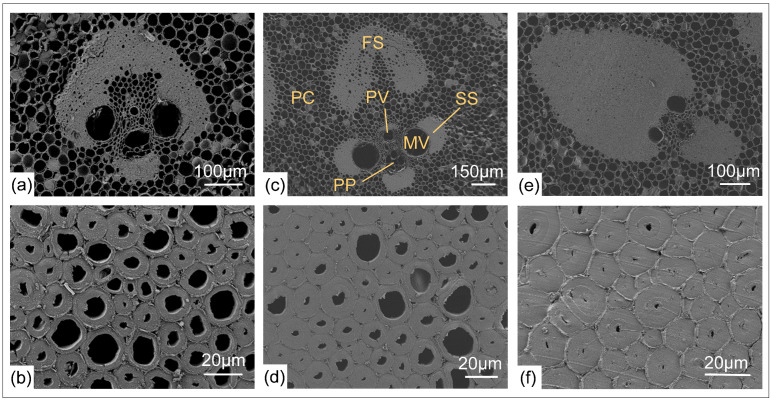
SEM images of *O. abyssinica*. (**a**): the inner section’s vascular bundle; (**b**): the inner fibers; (**c**): the middle section’s vascular bundle (FS: fiber strand; MV: metaxylem vessel; PV: protoxylem vessel; PC: parenchyma; PP: primary phloem; SS: sclerenchyma sheath); (**d**): the middle fibers; (**e**): the outer section’s vascular bundle; (**f**): the outer fibers.

**Figure 4 plants-12-02987-f004:**
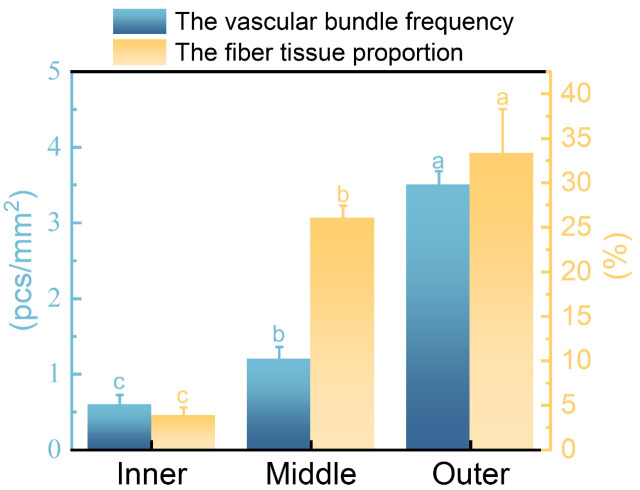
The frequency of vascular bundle and the tissue proportion of fiber. Different letters indicate significant differences among different radial regions (*p* < 0.05).

**Figure 5 plants-12-02987-f005:**
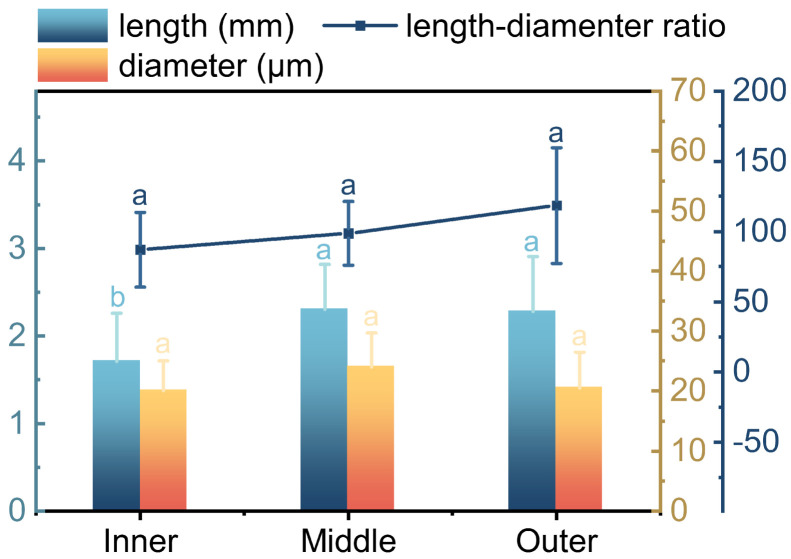
The dimension of fibers in *O. abyssinica*. Different letters indicate significant differences among different radial regions (*p* < 0.05).

**Figure 6 plants-12-02987-f006:**
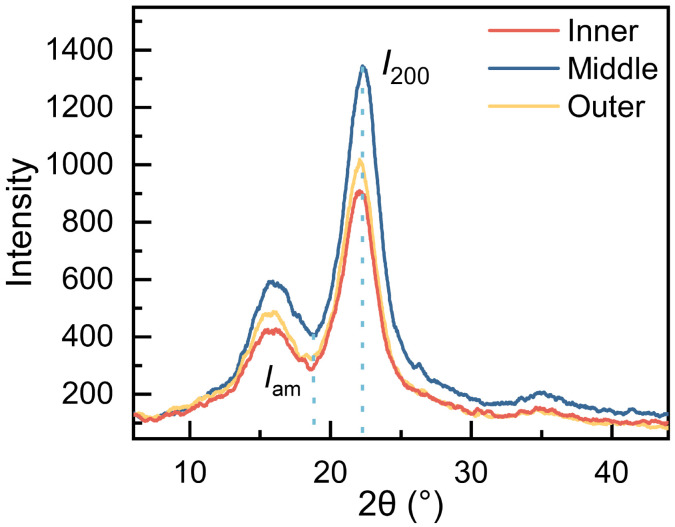
Determination of cellulose crystallinity of *O. abyssinica* fibers from X-ray diffraction.

**Figure 7 plants-12-02987-f007:**
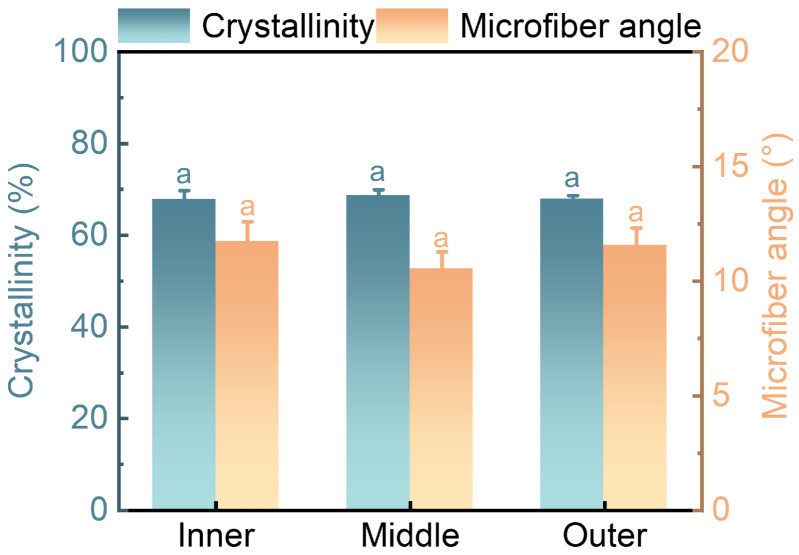
The cellulose crystallinity and MFA of fibers in *O. abyssinica*. Different letters indicate significant differences among different radial regions (*p* < 0.05).

**Figure 8 plants-12-02987-f008:**
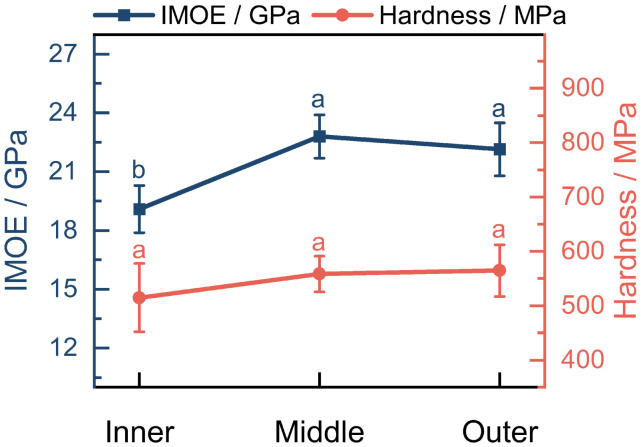
Indentation modulus of elasticity and hardness of cell walls of fibers in *O. abyssinica*. Different letters indicate significant differences among different radial regions (*p* < 0.05).

## Data Availability

The datasets generated during and/or analyzed during the current study are available from the corresponding author on reasonable request.

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
