# Peer review of "Fiber Characteristics and Mechanical Properties of Oxytenanthera abyssinica"

_plants, 2023, doi:10.3390/plants12162987_

Round 1

Reviewer 1 Report

I reviewed article entitled “Fiber Characteristics and Mechanical Properties of Oxytenanthera Abyssinica” by Genlin Tian et al. This is a straight forward experimental article which falls within the scope of your journal. I have following questions and comments for the authors.

1. I would expect to see the significant experimental values in the abstract section.

2.  In Materials and Methods, the experimental procedures were not corresponding to any International standard such as ASTM. How many replications do the tests were performed for each test? An ANOVA analysis was conducted without the hypothesis. In addition, the control should be included for the significant comparison.

3.  In Results and Discussions, the appropriate number of experimental values should be shown in corporate with an explanation such as in line 196-198 and so on.

4.  How to conclude Line 235-237, I didn’t see any experimental data to support this concluding remark.

5.  The comparison on the hardness of middle and outer layer is missing. Again, the significant value should be expressed in Line 261-262 to have a better understanding.

6.  In Conclusions, it was mentioned that the mechanical properties of O. abyssinica such as Young’s modulus, tensile strength, flexural properties, and crush resistance comparing to other bamboo or plant fibers are one of the key areas that requires further investigation. But I think it should be included here to strengthen the novelty of this research. An experimental design is necessary to polish the result of this research.

Author Response

Dear Reviewer,

Thank you for your consideration of our manuscript entitled “Fiber Characteristics and Mechanical Properties of Oxytenanthera Abyssinica”. We have addressed the reviewers’ comments point-by-point and revised the manuscript, all revisions have been clearly stated in the letter.

Thank you very much for your time and patience.

Best regards

Genlin Tian, Ph.D.

International Center for Bamboo and Rattan, Beijing, China

Reviewer #1 Comments and Suggestions for Authors

I reviewed article entitled “Fiber Characteristics and Mechanical Properties of Oxytenanthera Abyssinica” by Genlin Tian et al. This is a straight forward experimental article which falls within the scope of your journal. I have following questions and comments for the authors.

  1. I would expect to see the significant experimental values in the abstract section.

Response: Thank you for your valuable comments.We have added the necessary data to the abstract, and described the changes for each point.

We have changed “Results showed that the frequency of vascular bundles and the tissue proportion of fibers increased gradually from inner to outer. ” to “Results showed that the mean values of vascular bundle frequency and fiber tissue proportion were 1.76 pcs/m^2 and 21.04 %, respectively, and both increased gradually from inner to outer.”.

We have added “The mean length, diameter, and length-diameter ratio of the fiber were 0.21 mm, 21.54$\mu$m, and 101.41 respectively.”

We have removed “, while there was no significant difference in length between the middle- and outer fibers”.

We have added “The mean indentation modulus of elasticity (IMOE) and hardness were 21.34 Gpa and 545.88 Mpa. ”.

We have added “The mean crystallinity and microfibril angle(MFA) of the fibers was 68.12% and 11.26 degrees, respectively.”.

  1.  In Materials and Methods, the experimental procedures were not corresponding to any International standard such as ASTM. How many replications do the tests were performed for each test? An ANOVA analysis was conducted without the hypothesis. In addition, the control should be included for the significant comparison.

Response: Thank you for your valuable comments.We have not found any national standards for reference. We have added “The samples surfaces were scanned before and after testing to select valid nanoindentation test data for analysis of the results, where three samples had more than 30 valid data”; Each parameter, including fiber size, crystallinity, and MFA, was determined using more than 30 valid data points. We have added “ more than 30 measurements per data set.”.The original assumption is that there is no significant difference in mechanical properties between fibers in different parts (if there is a significant difference, reject the original assumption; if there is no significant difference, accept the original assumption)

  1.  In Results and Discussions, the appropriate number of experimental values should be shown in corporate with an explanation such as in line 196-198 and so on.

Response: Thank you for your valuable comments.We have incorporated intuitive data into the sentence “O. abyssinica (2.1 mm), Phyllostachys edulis (2.0 mm), Bambusa emeiensis (2.2 mm), Dendrocalamus giganteus Munro (2.7 mm).”;

We have added significant annotations to all images, “Different letters indicate significant differences among different radial regions (P<0.05).”

  1.  How to conclude Line 235-237, I didn’t see any experimental data to support this concluding remark.

Response: Thank you for your valuable comments.We have added the reference.[1] “Poletto, M.; Ornaghi Junior, H.L.; Zattera, A.J. Native cellulose: structure, characterization and thermal properties. Materials 2014, 7, 6105-6119.”

  1.  The comparison on the hardness of middle and outer layer is missing. Again, the significant value should be expressed in Line 261-262 to have a better understanding.

Response: Thank you for your valuable comments.We have added a discussion on the hardness of the middle and outer layers “The hardness of the inner region was 3.71 GPa lower than that of the middle region, while the hardness of the outer region was 0.65 lower than that of the middle region. The middle region exhibited the greatest fiber hardness, but there were only minor fluctuations in hardness along the radial direction, which did not result in significant differences.”;

We have added corresponding data “The mean IMOE and hardness of fibers in O. abyssinica was 21.34 GPa and 545.88 MPa, respectively. The cell walls of O. abyssinica exhibit superior mechanical properties that are comparable to the widely used fibers of Phyllostachys edulis (21.7 GPa and 610.00 MPa) but demonstrate superior strength when compared to Dendrocalamus farinosus (18.56 GPa and 410.72 MPa) [53 ,54 ]. This indicates the fibers of O. abyssinica may have a higher resistance to deformation and fracture than Dendrocalamus farinosus.”.

  1.  In Conclusions, it was mentioned that the mechanical properties of O. abyssinica such as Young’s modulus, tensile strength, flexural properties, and crush resistance comparing to other bamboo or plant fibers are one of the key areas that requires further investigation. But I think it should be included here to strengthen the novelty of this research. An experimental design is necessary to polish the result of this research.

However it is important to highlight the conclusion with more bullet points and summarize the main conclusion also in the abstract.

Response: Thank you for your valuable comments. We agree with your opinion, this proposal is valuable and very necessary for further understanding of solid bamboo. However, the topic of this paper is about the morphological characteristics and cell wall mechanics of solid bamboo fibers. Due to the limitations of materials and other factors, the current research on fiber mechanics belongs to the micro-scale. We will further discuss and study the macro-mechanics of solid bamboo in the future. This is an English translation of your sentence.

the quality of the pictures could also be further enhanced.

Response: Thank you for your valuable comments. We have improved the quality of the images.

please have the paper reviewed by a native English editor for minor English editing.

Response: Thank you for your valuable comments.We've had a native English editor read the article and corrected it for errors according to his comments.

Reviewer 2 Report

The manuscript titled “Fiber Characteristics and Mechanical Properties of Oxytenanthera Abysssinica” by Yu, L.; et al. is an original research work where the authors study the morphology, cellulose crystallinity and nanomechanical properties of bamboo fibers by complementary techniques as scanning electron microscopy (SEM), X-ray diffraction (XRD) and nanoindentation measurements. The authors found an increase of the mechanical properties according the cellulose crystallinity and the contrary effect regarding the bambo fibril angle. The study is interesting and it is well-designed.

However, it exists some points that need to be addressed (please, see them below detailed point-by-point). The most relevant outcomes found by the authors can contribute in the growth of many Industrial fields like food packaging, fabrication of composite materials, or papermaking, among others. For this reason, I will recommend the present scientific manuscript for further publication in Plants once all the below described suggestions will be properly fixed.

Here, there exists some points that must be covered in order to improve the scientific quality of the manuscript paper:

1) ABSTRACT. (OPTIONAL). The authors have added the term “cellulose crystallinity” in the Keyword list but no explanation is provided about the most relevant outcomes found by the authors related to “cellulose crystallinity” in the Abstract section. Maybe the authors should consider to add a small statement in this regard.

2) INTRODUCTION. “Bambo also represents a renewable (…) variety of applications, such as construction, furniture, papermaking, and so on” (lines 16-22). Here, the authors provide some insights about the advantages shown by bamboo fibers compared to other plant fibers. In this context, it may be desirable to funish some quantitative data to highlight the potential use of bamboo as greenfriendly material [1] like the shorter bamboo growth cycle  (from 3 to 5 years) respecting other plant fibers and the three times-fold greater carbon absorption capability (3.7 m3 of CO2 per hectare and day) compared to other plants.

[1] Borowski, P.F.; et al. Innovative Industrial Use of Bamboo as Key “Green” Material. Sustainability 2022, 14, 1955. https://doi.org/10.3390/su14041955.

3) MATERIALS AND METHODS. “Healthy four-year-old (…) were randomly selected” (lines 72-73). How many bamboo fibers were collected in all? What is the population size used in this study?

4) Figure 1 (line 76). This image comes from any technique or is a schematic representation. In the second case, the authors should change the current figure caption by “Representative schamatic representation of (…)”.

5) “(…) using a E-1010 sputter coater (…)” (line 80). Please, the authors should add the name and country of the manufactuter. This comment should be taken into account for the rest of this section for all consumables, chemical reagents and techniques used in this work.

6) “The cross-sectional areas (…) at a resolution of 9600 ppi” (lines 85-88). Please, the authors should define the term “pixels per inch”. Then the abbreviation “ppi” should be placed between brackets.

7) “The frequency of vascular bundles (…) with Image J software” (lines 88-89). Please, the authors should cite the following reference [2]:

[2] Schneider, C.A.; et al. NIH Image to ImageJ: 25 years of image analysis. Nat. Methods 2012, 9, 671-675. https://doi.org/10.1038/nmeth.2089.

8) “2.5. X-ray diffraction (XRD)” (lines 102-127). What was the software tool employed by the authors to process the raw data obtained by this technique. Please, this information should be provided.

9) “Nanoindentation test” (lines 128-147). Why did the authors use a Triboindenter as the test apparatus instead of alternative single molecule techniques [3]? The authors should add a briefly discussion about the convenience to use the Triboindenter as this tool is more cost-efficient and the data analysis less time-consuming than the aforementioned techniques.

[3] Magazzù, A.; et al. Investigation of Soft Matter Nanomechanics by Atomic Force Microscopy and Optical Tweezers: A comprehensive Review. Nanomaterials 2023, 13, 963. https://doi.org/10.3390/nano13060963.

10) RESULTS AND DISCUSSIONS. (OPTIONAL). The authors should consider to change the current title of this section by “Results and Discussion”.

11) Did the authors chemically characterized the bamboo fibers used in this study (for example by X-ray photoelectron spectroscopy, XPS or fourier transfor infrared spectroscopy, FTIR)? This is the most critical point of this work because it is neccesary the knowledge regarding the constituent composition of the studied bamboo fibers. In case affirmative, the authors need to furnish this information. Otherwise, the authors should refer previous works where the cellulose (26.0-43.0 %), hemicellulose (20.5 %), lignin (21.0-31.0 %), pectin (negligible content), wax (negligible content), ash and moisture (negligible content) were determined [4].

[4] Zwawi, M. A Review on Natural Fiber Bio-Composites, Surface Modifications and Applications. Molecules 2021, 26, 404. https://doi.org/10.3390/molecules26020404.

12) Figure 4 (line 186). The authors should state the respective standard deviation (SD) bars for each measured condition (as in the Figure 5 and Figure 7, respectively).

13) Figure 5 (line 205). “After one-way ANOVA analysis, there was no significant difference in the diamater of the fibers along the radial direction” (lines 208-209). (OPTIONAL) The authors should consider to add the information related to the existance or not of significant differences in the Figure 5 by adding marks (typically asterisks). Same comment for Figure 7 (line 244).

14) “The indentation modulus of elasticity (…) were 19.08 GPa, 22.79 GPa, and 22.14 GPa, respectively. The hardness (…) were 514.79 MPa, 558.21 MPa, and 564.65 MPa” (lines 249-251). Please, the authors should add the respective ±SD values for each measurement. Then, the authors should also modify “were” by “was” for the indentation modulus of elasticity and hardness, respectively.

15) CONCLUSIONS. The authors clearly state the most relevant finding obtained in this work.

“These factors could have a significant impact on the performance and durability of biocomposites made from this material” (lines 308-309). The authors also need to discuss the key role of interfacial forces between the plant fiber filler and the matrix and their impact on composite properties. It is neccesary to point out that bamboo fibers are highly lignified (please, see the comment poured in the point 11) like analogous kenaf and jute fibers. Lignin is a hydrophobic polymer. Thus, the forces displayed between the aforementioned lignified fibers and polar matrices like polybutilene succinate (PBS) are poor [5]. On the other hand, lignified plant fibers exert large interfacial forces with non-polar matrices as polypropylene (PP). This is a pivotal point to better understand the potential performance of those composites made of the combination of bamboo fibers with matrices of different chemical nature.

[5] Marcuello, C.; Chabbert, B.; Berzin, F.; Bercu, B.N.; Molinari, M.; Aguié-Béghin, V. Influence of Surface Chemistry of Fiber and Lignocellulosic Materials on Adhesion Properties with Polybutylene Succinate at Nanoscale. Materials 2023, 16, 2440. https://doi.org/10.3390/ma16062440.

16) REFERENCES. The references are not in the proper format style of Plants. The journal name should appear in abbreviated form. The authors should take care of this point.

The authors should take care of some existing English typos. For it, they need to carefully check out the manuscript to cover this point.

Author Response

Dear Reviewer,

Thank you for your consideration of our manuscript entitled “Fiber Characteristics and Mechanical Properties of Oxytenanthera Abyssinica”. We have addressed the reviewers’ comments point-by-point and revised the manuscript, all revisions have been clearly stated in the letter.

Thank you very much for your time and patience.

Best regards

Genlin Tian, Ph.D.

International Center for Bamboo and Rattan, Beijing, China

Comments and Suggestions for Authors

The manuscript titled “Fiber Characteristics and Mechanical Properties of Oxytenanthera Abysssinica” by Yu, L.; et al. is an original research work where the authors study the morphology, cellulose crystallinity and nanomechanical properties of bamboo fibers by complementary techniques as scanning electron microscopy (SEM), X-ray diffraction (XRD) and nanoindentation measurements. The authors found an increase of the mechanical properties according the cellulose crystallinity and the contrary effect regarding the bambo fibril angle. The study is interesting and it is well-designed.

However, it exists some points that need to be addressed (please, see them below detailed point-by-point). The most relevant outcomes found by the authors can contribute in the growth of many Industrial fields like food packaging, fabrication of composite materials, or papermaking, among others. For this reason, I will recommend the present scientific manuscript for further publication in Plants once all the below described suggestions will be properly fixed.

Here, there exists some points that must be covered in order to improve the scientific quality of the manuscript paper:

1) ABSTRACT. (OPTIONAL). The authors have added the term “cellulose crystallinity” in the Keyword list but no explanation is provided about the most relevant outcomes found by the authors related to “cellulose crystallinity” in the Abstract section. Maybe the authors should consider to add a small statement in this regard.

Response: Thank you for your valuable comments.We have added “The mean crystallinity and microfibril angle(MFA) of the fibers was 68.12% and 11.26 degrees, respectively”.

  • “Bambo also represents a renewable (…) variety of applications, such as construction, furniture, papermaking, and so on” (lines 16-22). Here, the authors provide some insights about the advantages shown by bamboo fibers compared to other plant fibers. In this context, it may be desirable to funish some quantitative data to highlight the potential use of bamboo as greenfriendly material [1] like the shorter bamboo growth cycle  (from 3 to 5 years) respecting other plant fibers and the three times-fold greater carbon absorption capability (3.7 m3 of CO2 per hectare and day) compared to other plants.

[1] Borowski, P.F.; et al. Innovative Industrial Use of Bamboo as Key “Green” Material. Sustainability 2022, 14, 1955. https://doi.org/10.3390/su14041955.

Response: Thank you for your valuable comments.We have added “It also has a three-fold greater carbon absorption capability than other plants, absorbing up to 3.7 m$^3$ of CO$_2$ per hectare and day”,And we have added corresponding references.[1]Borowski, P.F.; Patuk, I.; Bandala, E.R. Innovative industrial use of bamboo as key “Green” material. Sustainability 2022, 14, 1955.

3) MATERIALS AND METHODS. “Healthy four-year-old (…) were randomly selected” (lines 72-73). How many bamboo fibers were collected in all? What is the population size used in this study?

Response: Thank you for your valuable comments.Three bamboos were selected;We have discussed the fibers of three bamboos together.We have added “(length 32.86 ± 2.84 cm, diameter 40.11 ± 1.80 mm)”.We have added “with 11.5% water content”.

4) Figure 1 (line 76). This image comes from any technique or is a schematic representation. In the second case, the authors should change the current figure caption by “Representative schamatic representation of (…)”.

Response: Thank you for your valuable comments. We have changed the current figure caption by “Representative schamatic representation of O. abyssinica”.

5) “(…) using a E-1010 sputter coater (…)” (line 80). Please, the authors should add the name and country of the manufactuter. This comment should be taken into account for the rest of this section for all consumables, chemical reagents and techniques used in this work.

Response: Thank you for your valuable comments.We have added the model and origin of the microwave oven “(G80W23CSP-Z, Galanz, Foshan, China)”. We have added the model and origin of the sputtering machine “(E-1010, Quorum Technologies, Emsworth, UK)”.We have added purchasing companies and addresses for reagents “o, Analytical grade glacial acetic acid and 30 % H2O2 were purchased from Nanjing Chemical Reagent Co., Ltd”. We have added the model and address of the oven “(DHG-9240A, Shanghai Yiheng Technology Co., Ltd., Shanghai, China)”.

6) “The cross-sectional areas (…) at a resolution of 9600 ppi” (lines 85-88). Please, the authors should define the term “pixels per inch”. Then the abbreviation “ppi” should be placed between brackets.

Response: Thank you for your valuable comments.We have changed the first occurrence of “ppi” to “pixels per inch”.

7) “The frequency of vascular bundles (…) with Image J software” (lines 88-89). Please, the authors should cite the following reference [2]:

[2] Schneider, C.A.; et al. NIH Image to ImageJ: 25 years of image analysis. Nat. Methods 2012, 9, 671-675. https://doi.org/10.1038/nmeth.2089.[2]

Response: Thank you for your valuable comments.We have added the references you provided. “Schneider, C.A.; Rasband, W.S.; Eliceiri, K.W. NIH Image to ImageJ: 25 years of image analysis. Nature methods 2012, 9, 671-675.https://doi.org/10.1038/nmeth.2089.”.

8) “2.5. X-ray diffraction (XRD)” (lines 102-127). What was the software tool employed by the authors to process the raw data obtained by this technique. Please, this information should be provided.

Response: Thank you for your valuable comments. We have added “The Origin 2018 software was used to process the obtained raw data using the aforementioned method.”.

9) “Nanoindentation test” (lines 128-147). Why did the authors use a Triboindenter as the test apparatus instead of alternative single molecule techniques [3]? The authors should add a briefly discussion about the convenience to use the Triboindenter as this tool is more cost-efficient and the data analysis less time-consuming than the aforementioned techniques.

[3] Magazzù, A.; et al. Investigation of Soft Matter Nanomechanics by Atomic Force Microscopy and Optical Tweezers: A comprehensive Review. Nanomaterials 2023, 13, 963. https://doi.org/10.3390/nano13060963.[3]

Response: Thank you for your valuable comments.We have added “A Triboindenter as the test apparatus has more cost-efficiency and less time-consuming data analysis compared to alternative single molecule techniques” and added the references. “Magazzù, A.; Marcuello, C. Investigation of soft matter nanomechanics by atomic force microscopy and optical tweezers: A comprehensive review. Nanomaterials 2023, 13, 963.”.

10) RESULTS AND DISCUSSIONS. (OPTIONAL). The authors should consider to change the current title of this section by “Results and Discussion”.

Response: Thank you for your valuable comments.We have replaced “Results and Discussions” with “Results and Discussion”

11) Did the authors chemically characterized the bamboo fibers used in this study (for example by X-ray photoelectron spectroscopy, XPS or fourier transfor infrared spectroscopy, FTIR)? This is the most critical point of this work because it is neccesary the knowledge regarding the constituent composition of the studied bamboo fibers. In case affirmative, the authors need to furnish this information. Otherwise, the authors should refer previous works where the cellulose (26.0-43.0 %), hemicellulose (20.5 %), lignin (21.0-31.0 %), pectin (negligible content), wax (negligible content), ash and moisture (negligible content) were determined [4].

[4] Zwawi, M. A Review on Natural Fiber Bio-Composites, Surface Modifications and Applications. Molecules 2021, 26, 404. https://doi.org/10.3390/molecules26020404.[4]

Response: Thank you for your valuable comments.We have added “The chemical composition of bamboo fiber to include cellulose (26.0-43.0%), hemicellulose (20.5%), lignin (21.0-31.0%), pectin (negligible content), wax (negligible content), ash and moisture (negligible content)” and added the reference “Zwawi, M. A review on natural fiber bio-composites, surface modifications and applications. molecules 2021, 26, 404.”.

12) Figure 4 (line 186). The authors should state the respective standard deviation (SD) bars for each measured condition (as in the Figure 5 and Figure 7, respectively).

Response: Thank you for your valuable comments.We have added error bars to the image.

13) Figure 5 (line 205). “After one-way ANOVA analysis, there was no significant difference in the diamater of the fibers along the radial direction” (lines 208-209). (OPTIONAL) The authors should consider to add the information related to the existance or not of significant differences in the Figure 5 by adding marks (typically asterisks). Same comment for Figure 7 (line 244).

Response: Thank you for your valuable comments.We have added significant annotations to all images, “Different letters indicate significant differences among different radial regions (P<0.05).”

14) “The indentation modulus of elasticity (…) were 19.08 GPa, 22.79 GPa, and 22.14 GPa, respectively. The hardness (…) were 514.79 MPa, 558.21 MPa, and 564.65 MPa” (lines 249-251). Please, the authors should add the respective ±SD values for each measurement. Then, the authors should also modify “were” by “was” for the indentation modulus of elasticity and hardness, respectively.

Response: Thank you for your valuable comments.We changed 'were' to 'was' and added standard deviation after the data.

15) CONCLUSIONS. The authors clearly state the most relevant finding obtained in this work.

“These factors could have a significant impact on the performance and durability of biocomposites made from this material” (lines 308-309). The authors also need to discuss the key role of interfacial forces between the plant fiber filler and the matrix and their impact on composite properties. It is neccesary to point out that bamboo fibers are highly lignified (please, see the comment poured in the point 11) like analogous kenaf and jute fibers. Lignin is a hydrophobic polymer. Thus, the forces displayed between the aforementioned lignified fibers and polar matrices like polybutilene succinate (PBS) are poor [5]. On the other hand, lignified plant fibers exert large interfacial forces with non-polar matrices as polypropylene (PP). This is a pivotal point to better understand the potential performance of those composites made of the combination of bamboo fibers with matrices of different chemical nature.

[5] Marcuello, C.; Chabbert, B.; Berzin, F.; Bercu, B.N.; Molinari, M.; Aguié-Béghin, V. Influence of Surface Chemistry of Fiber and Lignocellulosic Materials on Adhesion Properties with Polybutylene Succinate at Nanoscale. Materials 2023, 16, 2440. https://doi.org/10.3390/ma16062440.

Response: Thank you for your valuable comments. We have added “Interfacial adhesion mechanisms are among the most influential yet seldom discussed factors that affect the physical, mechanical, and thermal properties of plant-fiber-reinforced polymer composites. In particular, the chemical nature of the fibers and the matrix can greatly affect the interfacial adhesion between them. Bamboo fibers are highly lignified, similar to kenaf and jute fibers. Lignin is a hydrophobic polymer, which means that it does not interact well with polar matrices such as polybutylene succinate (PBS). On the other hand, lignified plant fibers can display strong interfacial forces with non-polar matrices such as polypropylene (PP).” to the last paragraph of the conclusion and added the reference.

16) REFERENCES. The references are not in the proper format style of Plants. The journal name should appear in abbreviated form. The authors should take care of this point.

Response: Thank you for your valuable comments.We have made modifications to the references in the format of plants

Comments on the Quality of English Language

The authors should take care of some existing English typos. For it, they need to carefully check out the manuscript to cover this point.

Response: Thank you for your valuable comments.We have scrutinized the article and corrected any errors that appeared in it.

Submission Date

05 July 2023

Date of this review

01 Aug 2023 13:55:41

Round 2

Reviewer 1 Report

All comments received have been appropriately addressed with justifications.

Reviewer 2 Report

The authors did a great effort in order to address all the suggestions. For it, the scientific quality of the manuscript has significantly improved. I warmly recommend this work for further publication in Plants based on its novelty and the scope covered by the journal.